# Factors associated with the ability of adolescent girls and young women (AGYW) in sexual unions to negotiate for safer sex. An analysis of data from the 2018 Zambia Demographic and Health Survey (ZDHS)

**Teebeny Zulu** [1] *, **Mwiche Musukuma**[1], **Choolwe Jacobs** [1,2], **Patrick Musonda**[1,3]

**1** Department of Epidemiology and Biostatistics, School of Public Health, University of Zambia, Lusaka, Zambia, **2** Women in Global Health, Lusaka, Zambia, **3** University of Bergen, Bergen, Norway

* teebenyzulu@gmail.com

## Abstract

The ability of AGYW to negotiate for safer sex is key in the fight against the Human Immuno-deficiency Virus (HIV). We determined the prevalence of safer sex negotiation among AGYW in sexual unions aged 15–24 and its associated factors in Zambia. Of 1879 respondents, 78.0% (1466) had the ability to negotiate for safer sex (ANSS). While adjusting for other variables in the model, condom use at last sex with the most recent partner (AOR 4.08, 95% CI 1.74–9.60, p = 0.001), experiencing any sexual violence by husband or partner (AOR 1.74, 95% CI 1.17–2.59, p = 0.006), listening to the radio at least once a week (AOR 2.03, 95% CI 1.32–3.13, p = 0.001), secondary or higher education (AOR1.77, 95% CI 1.04–2.99, p = 0.034), being in the richest wealth quintile (AOR 2.70, 95% CI 1.30–5.60, p = 0.008), and living in Eastern Province (AOR 2.75, 95% CI 1.53–4.93 p = 0.001), North-western (AOR 2.31, 95% CI 1.15–4.65, p = 0.019) and Southern (AOR 3.11, 95% CI 1.58–6.09, p = 0.001) was associated with a significant increase in the odds of ANSS among AGYW aged 15–24 years in sexual unions. On the other hand, being in Muchinga province (AOR 0.48, 95% CI 0.28–0.81, p = 0.006) decreased the odds of ANSS. In conclusion, safer sex negotiation is crucial in combating HIV; hence, tailor-made interventions that promote condom use, frequency of listening to health programmes on the radio, education, and wealth acquisition should be implemented to build and sustain safer sex negotiation, particularly among AGYW in sexual unions.

## 1. Introduction

"Women submit while men love" is a prevalent narrative that has deprived women of their right to ask for condom use or deny sex even in their husband's extramarital affairs. Such ideas, among others, have aided the spread of HIV, resulting in Acquired Immune Deficiency Syndrome (AIDS), which has remained a significant global health burden. Globally, around 39

**Data Availability Statement:** Third party data was obtained for this study from the DHS Program.

Data may be requested from the DHS Program after creating an account and submitting a concept note. More access information can be found on the DHS Program website (https://dhsprogram.com/data/Access-Instructions.cfm). Data for this study were analyzed from the 2018 Zambia Demographic and Health Survey (ZDHS). The authors confirm that interested researchers would be able to access these data in the same manner as the authors. The authors also confirm that they had no special access privileges that others would not have.

**Funding:** The authors received no specific funding for this work.

**Competing interests:** The authors have declared that no competing interest exist.

million people were living with HIV in 2022, with 1.3 million becoming newly infected [1]. In most parts of the world, AGYW are more vulnerable to HIV infection than their male counterparts [2]. In 2022, around 1.9 million AGYW worldwide were HIV positive, compared to 1.2 million of their male counterparts [2]. In Sub-Saharan Africa in 2022, AGYW comprised over 77% of newly reported infections in individuals aged 15 to 24 [3]. Moreover, young women in Sub-Saharan Africa aged 15 to 24 had over three times higher chances of acquiring HIV in 2022 compared to young men [3]. Among other reasons, early marriage reduces AGYW's ability to negotiate for safer sex which increase their risk of contracting HIV compared to their male peers [4]. To reduce the cases of HIV, efforts to improve safer sex negotiation in various populations, especially among AGYW in sexual unions and other sexually active populations, have intensified, but little has been achieved [5].

In Sub-Saharan African countries, such as Zambia, difficulties arise for AGYW in sexual relationships when it comes to safer sex negotiation, as men have more power and control in making sexual decisions, leading to higher HIV rates which is estimated at 5.7% compared to 1.8% among their male counterparts [6]. Given this information, SDG 5 emphasises the importance of gender equality in empowering AGYW to have more influence in reproductive matters, attitudes, and their capacity to negotiate for safer sex with their partners [7]. Literature has shown that safer sex negotiation is associated with comprehensive HIV/AIDS knowledge [8–10], maternal level of education [11–13], wealth and occupational status [14, 15], maternal age [13], mass media exposure (radio, television, and newspapers) [8, 16], religious affiliation [12, 17], and intimate partner's violence (IPV) and alcohol consumption [16, 18]. In Zambia, interventions such as comprehensive sexuality education, the promotion of condom use, and Determined, Resilience, Empowered, AIDS-Free, Mentored, and Safe (DREAMS), among others, aim to enhance AGYW's ability to have knowledge and control over sexuality issues [19]. Adolescent girls and young women's ANSS is crucial in reducing HIV incidence and prevalence as it reduces the likelihood of having unprotected sex.

Although the link between ANSS and decreasing rates of HIV and unwanted pregnancy is well known, there has been limited research on this phenomenon in Zambia. This study is therefore aimed at determining the prevalence and the factors associated with ANSS among AGYW in sexual unions aged 15–24 years by analysing data from the 2018 ZDHS.

## 2. Materials and methods

### 2.1 Study area and setting

The study analyzed data from all 10 provinces of Zambia, a southern African country that is part of the SSA. As of 2023, Zambia's population was 21 million, expected to grow at a rate of 2.8% per year [20]. Adolescents and young people aged 10–24 comprise 34.3% of the population [21]. Additionally, about 56% of the population lives in rural areas and depends mainly on subsistence agriculture [20].

### 2.2 Study design, data source and study population

This study was a cross-sectional study which constitutes a secondary analysis of microdata utilizing national-level data sourced from the ZDHS program. The ZDHS is a comprehensive, nationally representative household survey conducted by the Zambia Statistics Agency in collaboration with global partners, including ICF International and the United States Agency for International Development (USAID). The survey employs a two-stage sampling process, initially selecting enumeration areas (EAs) and subsequently households. All women in the 15–49 years age group who consented to take part in the survey and were usual members of the selected households or spent the night before the survey in the selected households were

interviewed. A sample of 13,683 women aged 15–49 and 12,132 men aged 15–59 in 12831 households were successfully interviewed [22]. For this specific study, we extracted all pertinent variables from the AGYW in sexual unions aged 15 to 24 years data files (individual recode) 2018 ZDHS datasets. Data was accessed 29[th] of April 2024. The authors in this study did not have access to information that could identify individual participants during or after data collection.

### 2.3 Study variables

The dependent variable was 'Ability to Negotiate for Safer Sex' (ANSS), characterised by two possible outcomes. This variable was created by composing two questions: 'Can you refuse sex with your partner?' and 'Can you ask your partner to use a condom during sex?' Those who responded 'yes' to one of the two questions were classified as having ANSS and given a '1 = yes' code; otherwise, '0 = no' [8]. A total of fifteen explanatory variables were analysed and categorised into socio-demographic and Behavioural characteristics. The socio-demographic variables comprised age, occupation, education level, wealth index, province, residence, and religious affiliation. Behavioural traits included comprehensive knowledge of HIV, frequency of exposure to newspapers, radio, and television, their partner's alcohol consumption, experiencing intimate partner violence (IPV), using condoms during their last sexual intercourse with their most recent partner, and the total number of current sexual partners. Comprehensive knowledge on HIV/AIDS was a composite variable'. It was defined as correctly knowing two ways to prevent HIV transmission and rejection of the three most common misconceptions about HIV. This variable was measured by asking each woman whether or not she agreed with the following five statements: (1) Consistent use of condoms during sexual intercourse can prevent HIV transmission; (2) Limiting sex to just one uninfected faithful partner can prevent HIV transmission; (3) A healthy-looking person can have HIV; (4) A person can get HIV through mosquito bites; (5) A person can get HIV by sharing food with an HIV-infected person. A respondent was considered to have comprehensive HIV/AIDS knowledge if he/she correctly responding to all the five questions and a code 'Yes' = 1 was assigned, otherwise a 'No' = 0 was given. The variables were not predetermined a priori but were chosen based on theoretical importance and practical significance for negotiating safer sex [8].

### 2.4 Operational definitions

In this study, ANSS was defined as being able to ask for condom use or refuse sex. Sexual union refers to being married or living with a partner in other words cohabiting. We further defined adolescent as a female in the age range of 10–19 while a young woman as a female in the age range from 20–30.

### 2.5 Statistical analysis

Frequencies and percentages were presented for descriptive statistics to measure the proportion of safer sex negotiation among respondents. Associations between ANSS and categorical predictors were explored using the chi-squared test if the assumptions were satisfied; otherwise, Fisher's exact test was used. To examine the association between quantitative predictors and the outcome, the Mann-Whitney test was used. Analytical analysis was performed using generalised linear models with a logit link. An investigator-led stepwise modelling approach was used at a 5% level of significance. Odds ratios have been presented as a measure of the association between a predictor and the outcome. Data was analysed in STATA version 16.0 (Stata Corp., College Station, TX, USA).

### 2.6 Ethical considerations

The ICF Institutional Review Board approved the 2018 ZDHS data survey protocols with the ICF Project Number: 132989.0.000.ZM. DHS.02. Permission to use the dataset was acquired from ICF Macro, and the dataset named ZMIR71DTA is available for download at https://www.dhsprogram.com/data. The user carefully adhered to the given guidelines, highlighting the sensitive nature of the information and the significance of not trying to uncover the identity of any household or individual surveyed for the study (maintaining anonymity). Approval was requested from the University of Zambia Biomedical Research Ethics Committee (UNZABREC) (Ref. Number 3459/2022 (Reference). Approval was also requested from the National Health Research Authority (NHRA) (Ref. Negative. NHRA008/03/07/2023)—(NHRA008 March 7, 2023).

## 3. Results

### 3.1 Characteristics of the study population

Results for the characteristics of the study population are presented in Tables 1 and 2. Out of 1879 AGYW in sexual unions aged 15–24 years interviewed in the 2018 ZDHS, 78.0% had ANSS. The majority (73.3%) lived in rural areas, and 64.0% were not working. Majority (93.3%) did not use condoms during their last sex with their most recent sexual partner, 86.8% did not experience any sexual violence from their partner; and 71.3% of their partners did not drink alcohol. The highest percentage (31.9%) belonged to the poorest wealth quintile, with only 7.2% in the richest wealth quintile. More than half (52.4%) attained primary education, and the majority (82.0%) were Protestants, with only 0.4% being Muslims. The majority (87.8%) did not read newspapers or magazines at all, compared to 1.4% who read almost every day and 60.9% who did not listen to the radio either.

### 3.2 Socio-demographic characteristics by ANSS

Table 1 also presents sample socio-demographic characteristics by ANSS. There was an association between the participant's age and ANSS (p = 0.0015). The highest prevalence (84.0%) of ANSS was observed in participants residing in urban areas (p<0.0001), suggesting a strong association. There was a strong positive association between the highest level of education acquired and ANSS (p<0.0001). The highest prevalence of ANSS (82.1%) was observed among those who attained secondary or higher education. Respondents who were in the richest wealth quintile had the highest prevalence (91.1%) of ANSS (p<0.0001). A strong association (p<0.0001) was also observed between the province in which a participant resided and ANSS. The highest prevalence (90.2%) of ANSS was observed in AGYW in the eastern province.

### 3.3 Behavioural characteristics by ANSS

Sample sexual behavioural characteristics by ANSS are presented in Table 2. From the results, 92.0% of AGYW who used condoms at last sex with their most recent partner had ANSS, (p<0.0001). Roughly, an increase in the frequency of listening to the radio was associated with an increase in the prevalence of ANSS among participants (p = 0.001). The frequency of watching television was strongly associated with ANSS (p<0.0001). The highest prevalence of ANSS (86.7%) was observed in participants who watched television almost every day.

**Table 1. Socio-demographic characteristics of AGYW aged 15–24 in Zambia stratified by ability to negotiate for safer sex and their associations five years preceding the 2018 ZDHS.**

| Factors | Overall | Ability to negotiate for safer sex N = 1879(100%) | | p-value |
|---|---|---|---|---|
| | | No N = 413(22.0%) | Yes N = 1466(78.0%) | |
| Respondent's age, *median (IQR)* | 21(20, 23) | 21(19, 23) | 21(20, 23) | 0.0015[M] |
| **Type of place of residence** | | | | |
| Urban | 501 (26.7%) | 80(16.0%) | 421(84.0%) | <0.0001[C] |
| Rural | 1,378 (73.3%) | 333(24.2%) | 1045(75.8%) | |
| **occupational status** | | | | |
| Not working | 1203 (64.0%) | 266 (22.1%) | 937(77.89%) | 0.854[C] |
| Working | 676 (36.0%) | 147 (21.8%) | 529(78.3%) | |
| **Wealth index** | | | | |
| Poorest | 597 (31.8%) | 155(26.0%) | 442(74.0%) | <0.0001[CT] |
| Poorer | 451 (24.0%) | 107(23.7%) | 344 (76.3%) | |
| Middle | 389 (20.7%) | 85(21.9%) 54 (17.6%) | 304 (78.2%) | |
| Rich | 307 (16.3%) | 12 (8.9%) | 253 (82.4%) | |
| Richest | 135 (7.2%) | | 123 (91.1%) | |
| **Highest level of education** | | | | |
| no education | 118 (6.3%) | 39(33.1%) | 79(67.0%) | <0.0001[CT] |
| primary | 984 (52.4%) | 235(23.9%) | 749(76.1%) | |
| secondary or higher | 777 (41.4%) | 139(17.9%) | 620(82.1%) | |
| **Religion** | | | | |
| Catholic | 316 (16.8%) | 81(25.6%) | 235(74.4%) | 0.074[F] |
| Protestant | 1541 (82.0%) | 324(21.0%) | 1217(79.0%) | |
| Muslim | 7 (0.4%) | 2(28.6%) | 5(71.4%) | |
| Other | 15 (0.8%) | 6(40.0%) | 9(60.0%) | |
| **Province** | | | | |
| Central | 189 (10.1%) | 47(24.9%) | 142(75.1%) | <0.0001[C] |
| Copperbelt | 161 (8.6%) | 44(27.3%) | 117(72.7%) | |
| Eastern | 307 (16.3%) | 30(9.8%) | 277(90.2%) | |
| Luapula | 185 (9.9%) | 56(30.3%) | 129(69.7%) | |
| Lusaka | 192 (10.2%) | 42(21.9%) | 150(78.1%) | |
| Muchinga | 190 (10.1%) | 69(36.3%) | 121(63.7%) | |
| Northern | 213 (11.3%) | 67(31.5%) | 146(68.4%) | |
| Northwestern | 128 (6.8%) | 16(12.5%) | 112(87.5%) | |
| Southern | 199 (10.6%) | 24(12.1%) | 175(88.0%) | |
| Western | 115 (6.1%) | 18(15.6%) | 97(84.4%) | |

M = Mann Whitney test; **C** = Chi-squared test; **CT** = Chi-squared for trend; **F** = Fisher's exact test; **IQR** = interquartile range

### 3.4 Factors associated with ANSS among AGYW in sexual unions aged 15–24 years

In bivariate logistic regression analysis as shown in Table 3, the following variables had a statistically significant effect on ANSS; condom use during last sex with the most recent partner increased the odds of ANSS by a factor of 3.42 (COR 3.2, 95% CI 1.76–6.55, p<0.0001), compared to no education, primary as well as secondary or higher education increased the odds of ANSS (COR 1.57, 95% CI 1.04–2.37, p = 0.030) and (COR 2.27, 95% CI 1.48–3.47, p<0.0001)

**Table 2. Behavioural characteristics of AGYW aged 15–24 in Zambia stratified by ability to negotiate for safer sex and their associations five years preceding the 2018 ZDHS.**

| Factors | Overall | Ability to negotiate for safer sex N = 1879(100%) | | p-value |
|---|---|---|---|---|
| | | **No** **N = 413(22.0%)** | **Yes** **N = 1466(78.0%)** | |
| **Number of sexual partners** *median (IQR)* | 2(1, 2) | 2(1, 2) | 2(1, 2) | 0.2362[M] |
| **Condom used during last sex** | | | | |
| *No* | 1733 (93.3%) | 398(23.0%) | 1,335(77.0%) | <0.0001[C] |
| *Yes* | 124 (6.7%) | 10(8.1%) | 114(92.0%) | |
| **Intimate partner violence** | | | | |
| No | 1400 (86.7%) | 318(22.7%) | 1082(77.3%) | 0.070[C] |
| Yes | 215 (13.3%) | 37(17.2%) | 178(82.8%) | |
| **Husband/partner drinks alcohol** | | | | |
| No | 1151 (71.3%) | 243(21.1%) | 908(78.9%) | 0.0184[C] |
| Yes | 464 (28.7%) | 112(24.1%) | 352(75.9%) | |
| **Comprehensive HIV knowledge** | | | | |
| No | 1073 (59.1%) | 236(22.0%) | 837(78.0%) | 0.774[C] |
| Yes | 742 (40.9%) | 159(21.4%) | 583(78.6%) | |
| **Frequency of reading newspaper or magazine** | | | | |
| *Not at all* | 1650 (87.8%) | 366(22.2%) | 1284(77.8%) | 0.903[CT] |
| *Less than once a week* | 124 (6.6%) | 27(21.8%) | 97(78.2%) | |
| *At least once a week* | 79 (4.2%) | 15(19.0%) | 64(81.0%) | |
| *Almost every day* | 26 (1.4%) | 5(19.2%) | 21(80.8%) | |
| **Frequency of listening to radio** | | | | |
| *Not at all* | 1144 (60.9%) | 278(24.3%) | 866(75.7%) | 0.001[CT] |
| *Less than once a week* | 198 (10.5%) | 48(24.2%) | 150(75.8%) | |
| *At least once a week* | 257 (13.7%) | 35(13.6%) | 222(86.4%) | |
| *Almost every day* | 280 (14.9%) | 52(18.6%) | 228(81.4%) | |
| **Frequency of watching television** | | | | |
| *Not at all* | 1441 (76.7%) | 328(23.5%) | 1103(76.5%) | <0.0001[CT] |
| *Less than once a week* | 80 (4.3%) | 22(27.5%) | 58(72.5%) | |
| *At least once a week* | 88 (4.7% | 17(19.3%) | 71(80.7%) | |
| *Almost every day* | 270 (14.4%) | 36(13.3%) | 234(86.7%) | |

**M** = Mann Whitney test; **C** = Chi-squared test; **CT** = Chi-squared for trend; **IQR** = interquartile range.

Three-quarters (76.7%) did not watch television at all, with only 14.4% watching almost every day. The highest proportion of the participants (16.3%) came from the eastern province, while the western province contributed the least (6.1%). Less than half of them (40.9%) had comprehensive HIV knowledge.

respectively, being in the richer wealth quintile compared to being in the poorest, increased the odds of ANSS by 35% (COR 1.64, 95% CI 1.16–2.32, p = 0.005), compared to being in the poorest wealth quintile, being in the richest increased the odds of ANSS by 3.60 times (COR 3.60, 95% CI 1.93–6.68, p<0.0001), the odds of ANSS among those who listened to the radio at least once a week were 2.04 times the odds of those who did not listen at all (COR 2.04, 95% CI 1.39–2.98, p<0.0001), compared to not listening to the radio at all, listening to the radio almost every day increased the odds of ANSS by 41% (COR 1.41, 95% CI 1.01–1.96, p = 0.042), as compared to being in central province, being in eastern province increased the odds of ANSS by 2.75 times (COR 2.75, 95% CI: 1.53–4.93, p = 0.001), this was the case with those in North-western and Southern province (COR 2.31, 95% CI 1.15–4.65, p = 0.019) and (COR 3.11, 95%

**Table 3. Unadjusted and adjusted odds ratios for predictors of the ability to negotiate for safer sex among AGYW aged 15–24 in Zambia five years preceding the 2018 ZDHS.**

| Factor | UOR (95% CI) | p-value | AOR (95% CI) | p-value |
|---|---|---|---|---|
| **Condom used during last sex** | | | | |
| *No* | Ref | n/a | Ref | n/a |
| *Yes* | 3.40 (1.76, 6.55) | <0.0001 | 4.08 (1.74, 9.60) | 0.001 |
| **Intimate partner violence** | | | | |
| *No* | Ref | n/a | Ref | n/a |
| *Yes* | 1.41 (0.97, 2.06) | 0.071 | 1.74 (1.17, 2.59) | 0.006 |
| **Highest level of education** | | | | |
| *No education* | Ref | n/a | Ref | n/a |
| *Primary* | 1.57 (1.04, 2.37) | 0.030 | 1.33 (0.82, 2.15) | 0.254 |
| *Secondary or higher* | 2.27 (1.48, 3.47) | <0.0001 | 1.77 (1.04, 2.99) | 0.034 |
| **Frequency of listening to radio** | | | | |
| *Not at all* | Ref | n/a | Ref | n/a |
| *Less than once a week* | 1.00 (0.71, 1.43) | 0.986 | 0.91 (0.61, 1.34) | 0.629 |
| *At least once a week* | 2.04 (1.39, 2.98) | <0.0001 | 2.03 (1.32, 3.13) 1.30 (0.88, 1.92) | 0.001 |
| *Almost every day* | 1.41 (1.01, 1.96) | 0.042 | | 0.180 |
| **Wealth index** | | | | |
| *Poorest* | Ref | n/a | Ref | n/a |
| *Poorer* | 1.13 (0.85, 1.50) | 0.408 | 0.97 (0.69, 1.35) | 0.845 |
| *Middle* | 1.25 (0.93 1.70) | 0.142 | 1.06 (0.72, 1.54) | 0.775 |
| *Richer* | 1.64 (1.16, 2.32) | 0.005 | 1.37 (0.86, 2.19) | 0.190 |
| *Richest* | 3.60 (1.93, 6.68) | <0.0001 | 2.70 (1.30, 5.60) | 0.008 |
| **Province** | | | | |
| *Central* | Ref | n/a | Ref | n/a |
| *Copperbelt* | 0.88 (0.55, 1.42) | 0.601 | 0.65 (0.37, 1.14) | 0.135 |
| *Eastern* | 3.06 (1.85, 5.04) | <0.0001 | 2.75 (1.53, 4.93) | 0.001 |
| *Luapula* | 0.76 (0.48, 1.20) | 0.243 | 0.69 (0.40, 1.18) | 0.172 |
| *Lusaka* | 1.18 (0.73, 1.90) | 0.490 | 0.76 (0.43, 1.36) | 0.359 |
| *Muchinga* | 0.58 (0.37, 0.90) | 0.016 | 0.48 (0.28, 0.81) | 0.006 |
| *Northern* | 0.72 (0.47, 1.11) | 0.144 | 0.73 (0.43, 1.22) | 0.229 |
| *Northwestern* | 2.32 (1.25, 4.30) | 0.008 | 2.31 (1.15, 4.65) | 0.019 |
| *Southern* | 2.41 (1.41, 4.13) | 0.001 | 3.11 (1.58, 6.09) | 0.001 |
| *Western* | 1.78 (0.98, 3.25) | 0.059 | 1.84 (0.93, 3.67) | 0.081 |

**UOR** = unadjusted odds ratio, **AOR** = adjusted odds ratios from complete case analysis; **Ref** = reference category; **n/a** = not applicable; **CI** = confidence interval

CI 1.58–6.09, p = 0.001). Being in Muchinga province decreased the odds of ANSS by 52% (COR 0.48, 95% CI 0.28–0.81, p = 0.006).

Adjusted multiple logistic regression results are also presented in Table 3. While controlling for other variables in the model; the odds of ANSS among AGYW who used condoms during their last sex with the most recent partner were 4.08 times the odds of those who did not use condoms (AOR 4.08, 95% CI 1.74–9.60, p = 0.001), as compared to AGYW in sexual unions who did not experience IPV, the odds of ANSS among AGYW in sexual unions who experienced IPV, increased by 74% (AOR 1.74, 95% CI 1.17–2.59, p = 0.006). The odds of ANSS among AGYW in sexual unions whose highest level of education was secondary or higher were 1.77 times the odds of their counterparts with no education (AOR 1.77, 95% CI 1.04–2.99, p = 0.034), listening to the radio at least once a week as compared to not listening to the

radio at all, increased the odds of ANSS among AGYW by a factor of 2.03 (AOR 2.02, 95% CI 1.32–3.13, p<0.0001), being in the richest wealth quintile as compared to being in the poorest among AGYW increased the odds of ANSS by 2.70 times (AOR 2.70, 95% CI 1.30–5.60, p = 0.008), the odds of ANSS among AGYW in Eastern province were 2.75 times the odds of AGYW in sexual unions in central province of Zambia (AOR 2.75, 95% CI 1.53–4.90, p<0.0001), compared to AGYW in central province, being in Northwestern province increased the odds of ANSS by a factor of 2.31 (AOR 2.31, 95% CI 1.15–4.65, p = 0.019), being in southern province as compared to being in the central province increased the odds of ANSS among AGYW by a factor of 3.11 (AOR 3.11, 95% CI 1.58–6.09, p = 0.001).

## 4. Discussion

In this study, we explored the prevalence and factors associated with ANSS among AGYW in sexual unions aged 15–24 years. The study revealed that the prevalence of ANSS among AGYW was high in Zambia five years before the 2018 ZDHS. Condom use during last sex with the most recent partner, experiencing any form of IPV, the highest level of education attained, frequency of listening to the radio, wealth index, and province were factors associated with ANSS among AGYW in sexual unions aged 15–24 years in Zambia.

The study found a high prevalence of safer sex negotiation in Zambia, which aligns with findings from previous studies conducted in Ethiopia [11] and SSA [8]. The reason for this may be the educational background of most participants in this study, as the majority had finished elementary school. Educated young women are more likely than their uneducated counterparts to negotiate for safer sex [11]. Formal education gives women correct information about STIs and HIV/AIDS, which likely improves their attitudes towards practising safe sex [23]. This finding implies that the government and other appropriate parties should introduce measures like covering tuition fees, exam fees, and uniforms to enhance education access, thereby building and sustaining AGYW's ANSS in Zambia [24].

Findings have indicated that AGYW in sexual unions who used condoms during their last sex with the most recent partner were more likely to negotiate for safer sex compared to those who did not use condoms. A prior study in SSA across 30 countries showed that most AGYW in sexual relationships who used condoms in their last sexual encounter were successful in negotiating for safer sex [8]. One possible explanation for this is that AGYW who consistently use condoms may be more familiar with them and are thus more inclined to dispel misconceptions about their use. Despite knowing that condoms offer protection against HIV/AIDS and early pregnancy, most men choose not to use them due to the belief that condoms diminish sexual pleasure and signify a lack of trust in a committed relationship [25]. Hence, by consistently using condoms, AGYW can dispel myths about condom use within sexual relationships and effectively advocate for safer sex. Therefore, it is important to continue encouraging the use of condoms among AGYW in sexual relationships through various platforms, including social media and sports, as they are heavily involved in these activities.

Similar to Melenedz and his team [26], our study found that AGYW who had experienced any form of IPV were more likely to participate in discussions about safer sex compared to those who had not. Many AGYW in sexual unions where sexual abuse occurs typically create ways to avoid being abused by their partners again. These women are more likely to insist on using condoms or reject sex if their partner wants to have unprotected sex. Another potential reason could be that when AGYW attempted to talk about practising safe sex, their partners may have perceived it as disrespectful and resorted to IPV. Previous studies have shown that women who have autonomy over their sexuality are more likely to face different forms of violence [16, 18]. This suggests that these women might be capable of standing up for their rights,

a situation their husbands see as a challenge to their authority, causing them to act aggressively towards their wives. Many African communities, especially in rural areas that prioritize traditional practices, prohibit women from refusing sexual advances from their partners unless they are menstruating, pregnant, breastfeeding, or engaged [16]. Our findings show that creating and improving a support system, such as 360 Women, is crucial for both IPV survivors and non-survivors to share ideas and offer each other support [16].

Our results indicate that AGYW who had completed secondary school or higher are more inclined to engage in safer sex negotiations compared to those with no education. The correlation identified in this research aligns with previous studies [12, 27]. Formal education boosts women's independence and ability to make decisions about their reproductive health and overall well-being [27]. Another possible reason for this finding may be that women who receive formal education acquire reliable information about STIs and HIV/AIDS, which likely enhances their willingness to discuss safer sex [23]. Studies indicate that women who are educated or have higher levels of education are more capable and empowered to alter their sexual behaviour and embrace safer sexual practices [28, 29]. Evidence shows that educated women or individuals with higher levels of education are better empowered to change their sexual behaviour and adopt safer sexual practices [27]. We recommend prioritizing the promotion of access and equity in education as a crucial element for effectively responding to HIV prevention among AGYW in sexual partnerships [24, 30].

Additionally, our research revealed that AGYW who tuned in to the radio at least once per week were more likely to negotiate for safer sex than those who never listened to the radio. Our results align with another study's results [16], which found a higher likelihood of safer sex negotiation among AGYW exposed to the mass media however, this study did not specify the genre of mass media. Research has indicated that AGYW exposed to the media acquire knowledge about HIV and are more likely to negotiate for safer sex, however, this was not a significant factor in our study [8]. Extensive information about HIV is frequently cited as a factor that impacts individuals' sexual practices and control over their own sexual experiences [31, 32]. There is a need to think about using radio to spread health messages to the public, promoting awareness and encouraging safe sex conversations.

In addition, we observed a higher chance of AGYW in the richest wealth quintile engaging in safer sex negotiations compared to those in the poorest wealth quintile. Previous studies also found a higher likelihood of safer sex negotiation among women who belonged to the middle class and richer wealth status [33, 34]. This is because AGYW from wealthier families have more financial security, which lowers the likelihood of them participating in transactional sex or suffering from abusive sexual relationships. Furthermore, these teenagers have access to good education and media in urban areas where working-class women serve as role models, motivating them to decrease their reliance on husbands. Many women who are financially reliant on men find it challenging to prioritise safe sexual practices, despite having a good understanding of HIV/AIDS [35]. This highlights the importance of implementing interventions aimed at enhancing the capacity of AGYW in sexual unions to negotiate for safer sex, specifically focusing on poverty alleviation initiatives in low-income households.

It has been shown in this study that AGYW in sexual unions who resided in Eastern, Northwestern, or southern provinces were more likely to negotiate for safer sex as compared to their counterparts in Central Province. On the other hand, AGYW in Muchinga province had a reduced ability to negotiate for safer sex. This could be explained by differences in programmes and levels of sensitization to safer sex practices in different provinces [22]. Qualitative studies are needed to understand why the Eastern, Northwestern, and Southern provinces are doing better concerning safer sex negotiation.

### 4.1 Study strengths and limitations

The main strength of this study lies in its utilisation of a sample that is representative of the entire nation, with collection techniques and methodology that adhere to the highest standards. Therefore, the results of the research can apply to all AGYW aged 15 to 24 in sexual unions in Zambia. Among the limitations of this study is the use of a cross-sectional design, preventing the determination of causality. Furthermore, the connections between the independent and dependent variables may change with time, indicating the necessity for research on this subject utilising the most recent ZDHS. Additionally, it is important to investigate the impact of social media platforms like Facebook and TikTok on the negotiation of safer sex.

### 4.2 Conclusion

We determined the prevalence and factors influencing the ability to negotiate for safer sex among AGYW aged 15–24 in Zambia. A significant number of AGYW had the ability to negotiate for safer sex five years before the 2018 ZDHS. Factors such as condom use, IPV, frequency of listening to the radio, having a secondary education or higher, richest wealth quintile, and residing in the Eastern, Northwestern, and Southern provinces were all linked to a higher chance of negotiating safer sex. Conversely, living in Muchinga Province was linked to a decreased likelihood of engaging in safer sex negotiations. As a result, it is necessary to introduce measures to improve access to education, wealth, condom usage, and mass media to build and sustain safer sex negotiations among AGYW in Zambia. Additional research, such as qualitative research, may be required to understand why AGYW in sexual relationships in the Eastern, Northwestern, and Southern provinces are successful in negotiating safer sex.

## Author Contributions

**Conceptualization:** Teebeny Zulu, Choolwe Jacobs.

**Data curation:** Teebeny Zulu.

**Formal analysis:** Teebeny Zulu, Patrick Musonda.

**Methodology:** Teebeny Zulu.

**Project administration:** Teebeny Zulu.

**Resources:** Teebeny Zulu.

**Software:** Patrick Musonda.

**Supervision:** Mwiche Musukuma.

**Visualization:** Teebeny Zulu.

**Writing – original draft:** Teebeny Zulu.

**Writing – review & editing:** Choolwe Jacobs.

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
