## [Decision Letter · Decision Letter 0]

25 Jul 2024

PGPH-D-24-01136

Factors Associated with the Ability of Adolescent Girls and Young Women (AGYW) in Sexual Unions to Negotiate for Safer Sex. An Analysis of Data from the 2018 Zambia Demographic and Health Survey (ZDHS)

Dear Dr. Zulu,

Thank you for submitting your manuscript to PLOS Global Public Health. After careful consideration, we feel that it has merit but does not fully meet PLOS Global Public Health’s publication criteria as it currently stands. Therefore, we invite you to submit a revised version of the manuscript that addresses the points raised during the review process.

Please submit your revised manuscript by 05 August 2024 If you will need more time than this to complete your revisions, please reply to this message or contact the journal office at globalpubhealth@plos.org. Please include the following items when submitting your revised manuscript:

We look forward to receiving your revised manuscript.

Kind regards,

Lucy Chimoyi, PhD

Academic Editor

Journal Requirements:

Additional Editor Comments (if provided):

The reviewers have raised pertinent questions on the work the authors have submitted for publication to PLOS Global Public Health. In addition to the reviewers questions, 

1. Are any of the AGYW in age-disparate relationships? if yes, what is the proportion and what is the mean age of the sexual partners

2. The captions for the tables are not comprehensive enough and cannot pass the "fall to the ground test" What is being presented in the tables? The authors are advised to please revise these captions to include the population, outcome, setting and time period

3. Table 2, is the term sexual behaviour or sexuality behaviour?

4. Table 1 included the place of residence variable, where a significant difference is observed between the two groups that make up the outcome. However, this variable is missing from the univariate and multivariate models. Can the authors explain this omission? Could it be that some of the provinces/regions where AGYW were less likely to negotiate sex are largely rural?

5. How was comprehensive HIV knowledge assessed?

6. Limitations are missing from the manuscript. Can the authors highlight some limitations?

7. Is there a variable that shows alcohol consumption by AGYW or inebriated sexual intercourse?

Reviewers' comments:

Reviewer's Responses to Questions

**Comments to the Author**

1. Does this manuscript meet PLOS Global Public Health’s publication criteria? Is the manuscript technically sound, and do the data support the conclusions? The manuscript must describe methodologically and ethically rigorous research with conclusions that are appropriately drawn based on the data presented.

Reviewer #1: Yes

Reviewer #2: Yes

2. Has the statistical analysis been performed appropriately and rigorously?

Reviewer #1: Yes

Reviewer #2: Yes

3. Have the authors made all data underlying the findings in their manuscript fully available (please refer to the Data Availability Statement at the start of the manuscript PDF file)?

Reviewer #1: Yes

Reviewer #2: Yes

4. Is the manuscript presented in an intelligible fashion and written in standard English?

Reviewer #1: Yes

Reviewer #2: Yes

5. Review Comments to the Author

Reviewer #1: 1. Title has ignored ''refusal to have sex' as a major variable

2. Abstract is clear and representative of the purpose, method, findings, conclusions and recommendations

3. Introduction: Line 33 should be ''to negotiate condom'' not to ''request''. ...and ..or abstain from sex.....''

: define ''young women''

: after line 40, there is need for statistics of HIV prevalence among AGYW in Zambia (national statistics)

4. Lines 51-54: Provide literature on the role of soft skills such as self-esteem; self concept pillars such as self-efficacy;

and other psychological contributors such as HIV risk perception

5. Lines 67-71: Indicate the study area and setting in terms of marital status because the age range is wide

6. Line 87: Show the Independent variable/s

7. Line 94: Knowledge of HIV , exposure to newspapers radio and TV, partners' alcohol consumption and IPV are not a

sexual behaviour, rather, it is a non sexual attribute of HIV infection susceptibility

The sexual behaviours are are condom use and number of sexual partners (line 96)

8. * Operational definitions: Include who an adolescent is according to this study and what stage of adolescence this study focused on

9. Line 103:What attributes were the frequencies and percentages used to measure

10. Line 125: Instead of ''most of them'', use, ''the majority''

11. Tables 1 and 2. Place the corresponding description for table one under table one. Do this for table two too

12. After line 311: After the conclusions, suggestion mitigation strategies for low negotiation for safe sex and abstinence from sex among AGYW

Reviewer #2: In general, the manuscript is well-written and timely, given how inability to negotiate for sex, especially in this age group, contributes to HIV infections

Introduction:

Line 38: For clarity, include "male counterpart."

Line 42/43: The main area of study is negotiating for safer sex, which is one aspect of sexual autonomy. For clarity, consider expounding on what aspect of sexual autonomy is impacted.

Study Area and Setting:

The justification for why we need this data for the Zambian population is not very clear. Consider including HIV prevalence in Zambia, particularly the prevalence in the AGYW age group. Additionally,since socioeconomic factors affect the main outcome consider discussing socioeconomic status, is there any linkage with residing in rural areas as this is a contributing factor to the inability to negotiate for safer sex practices.

Line 44 to 45: Little has been achieved in negotiating safer sex practices. Are there reasons for this? Link it to the need for understanding the factors associated with the inability to negotiate for safer sex.

Line 100 to 101: The operational definition of ANSS should be rephrased to make it clearer.

6. PLOS authors have the option to publish the peer review history of their article (what does this mean?). If published, this will include your full peer review and any attached files.

**Do you want your identity to be public for this peer review?** For information about this choice, including consent withdrawal, please see our Privacy Policy.

Reviewer #1: **Yes: **Loyce Kobusingye

Reviewer #2: **Yes: **Irene Mugenya

---

## [Editor Report · Decision Letter 1]

7 Aug 2024

PGPH-D-24-01136R1

Factors Associated with the Ability of Adolescent Girls and Young Women (AGYW) in Sexual Unions to Negotiate for Safer Sex. An Analysis of Data from the 2018 Zambia Demographic and HealthSurvey (ZDHS)

Dear Dr. Zulu,

Thank you for submitting your manuscript to PLOS Global Public Health. After careful consideration, we feel that it has merit but does not fully meet PLOS Global Public Health’s publication criteria as it currently stands. Therefore, we invite you to submit a revised version of the manuscript that addresses the points raised during the review process.

We look forward to receiving your revised manuscript.

Kind regards,

Lucy Chimoyi, PhD

Academic Editor

Journal Requirements:

Additional Editor Comments (if provided):

Dear Dr Zulu,

Thank you for responding to the questions raised by the reviewers. I have two comments that are yet to be addressed

1. In the response, the authors clarify that there were no age-disparate relationships but did not mention the mean age of the male sexual partners. This is important because an older age (>5 years ) has been known to impact the ability to negotiate sex for AGYWs.

2.In the table showing "Sexual characteristics...." and the paragraph showing these results, i wonder why variables such as watching TV, reading newspaper etc are included. Can the authors revise the titles of the caption and paragraph or create a separate table and paragraph for these results? It is not clear why exposure to TV, radio and newspapers can be considered as a sexual behaviour trait (this is mentioned in the methods section)

3. The response on how comprehensive HIV knowledge was assessed is not convincing. Was comprehensive HIV knowledge a stand alone variable in the DHS dataset or a composite variable generated from a series of questions in the data? If the latter, i would expect the authors to explain how this was arrived at in the methods section, similar to the ANSS definition.
---

## [Editor Report · Decision Letter 2]

9 Aug 2024

Factors Associated with the Ability of Adolescent Girls and Young Women (AGYW) in Sexual Unions to Negotiate for Safer Sex. An Analysis of Data from the 2018 Zambia Demographic and HealthSurvey (ZDHS)

PGPH-D-24-01136R2

Dear Dr Zulu,

We are pleased to inform you that your manuscript 'Factors Associated with the Ability of Adolescent Girls and Young Women (AGYW) in Sexual Unions to Negotiate for Safer Sex. An Analysis of Data from the 2018 Zambia Demographic and Health Survey (ZDHS)' has been provisionally accepted for publication in PLOS Global Public Health.

Best regards,

Lucy Chimoyi, PhD

Academic Editor